# Dietary Branched-Chain Amino Acids and Hyper-LDL-Cholesterolemia: A Case–Control Study Using Interpretable Machine-Learning Models in Chinese Children and Adolescents

**DOI:** 10.3390/nu17203280

**Published:** 2025-10-18

**Authors:** Zeping Zang, Shixiu Zhang, Changqing Liu, Yiya Liu, Meina Tian, Xiaoyan Luo, Qianrang Zhu, Lei Liu, Lianlong Yu

**Affiliations:** 1School of Public Health, Cheeloo College of Medicine, Shandong University, Jinan 250012, China; 202316375@mail.sdu.edu.cn (Z.Z.);; 2Shandong Center for Disease Control and Prevention, Jinan 250014, China; 3Hebei Center for Disease Control and Prevention, Shijiazhuang 050021, China; 4Guizhou Center for Disease Control and Prevention, Guiyang 550004, China; 5Jiangsu Provincial Center for Disease Control and Prevention, Nanjing 210009, China

**Keywords:** children and adolescents, BCAAs intake, hyper-LDL-cholesterolemia, cardiovascular disease

## Abstract

**Background**: Plasma branched-chain amino acid (BCAA) concentrations are positively associated with low-density lipoprotein cholesterol (LDL-C) levels. However, the relationship between dietary branched-chain amino acids and hyper-LDL-cholesterolemia is unclear in children and adolescents. **Methods**: This study explored the correlation between BCAAs and hyper-LDL-cholesterolemia risk through propensity score matching and conditional logistic regression. Machine learning based on LightGBM indicated the important role of BCAAs in the prediction of hyper-LDL-cholesterolemia. To examine the dose–response relationship, Restricted Cubic Splines (RCS) and receiver operating characteristic curves (ROC) were employed. The causal link between BCAA and cardiovascular disease (CVD) was explored via mediation Mendelian randomization. **Results**: For every 1 g/day increment in the intake of isoleucine, leucine, and valine, there was a corresponding 30%, 11%, and 16% rise in the risk of hyper-LDL-cholesterolemia, respectively. The optimal cut-off values stood at 5.53, 6.40, and 4.18 g/day, respectively. Utilizing the inverse variance weighted method for estimation revealed that the total effect of BCAA on CVD was OR = 1.06 (95% CI: 1.02~1.11), with *p* = 0.005. The indirect effect, mediated by LDL-C, was OR = 1.02 (95% CI: 1.00~1.02), with *p* = 0.026. The direct effect was noted at OR = 1.05 (95% CI: 1.01~1.09), with *p* = 0.017. **Conclusions**: Dietary BCAAs are positively correlated with hyper-LDL-cholesterolemia in children and adolescents. LDL-C serve as a mediator of CVD caused by BCAAs.

## 1. Introduction

Dyslipidemia in children and adolescents represents a critical public health challenge, with elevated low-density lipoprotein cholesterol serving as an important causal risk factor for atherosclerotic cardiovascular disease (ASCVD) [1]. In 2021, high low-density lipoprotein cholesterol was the seventh-leading Level 2 risk factor for deaths. It contributed to 3.65 million (2.13–5.26) deaths in 2021 [2]. Hyper-LDL-cholesterolemia in childhood and adolescence has garnered significant attention due to its potential to induce long-term impairments on cardiovascular health in adulthood [3,4].

BCAAs, including leucine, isoleucine, and valine, are primarily derived from animal-based proteins such as red meat [5]. While essential for growth and development, BCAAs play paradoxical roles in metabolic health. Elevated concentrations of BCAAs and associated metabolites are now broadly recognized as a metabolic hallmark of obesity, insulin resistance, and type 2 diabetes mellitus (T2DM) in humans [6,7]. Current research has demonstrated that serum BCAA contributes to the development of CVD [8,9,10]. On the other hand, BCAAs supplementation has been shown to prevent muscle loss and enhance muscle function; therefore, BCAAs are often used as a dietary supplement to assist in muscle protein synthesis [11,12]. An increased demand for BCAAs may result in increased consumption of foods such as red meat, which may bring more lipids. Therefore, the amount of BCAAs consumed is a concern, and it is necessary to ensure that the amount consumed is beneficial and not negative.

Although positive associations between BCAA and LDL-C have been observed in adults [13,14], the relationship of dietary BCAAs with LDL-C is unclear, especially in children and adolescents.

## 2. Materials and Methods

### 2.1. Study Design and Participants

The China National Nutrition and Health Surveillance of Children and Lactating Mothers is a multi-stage, stratified random sampling survey. However, due to dataset limitations, only data from four provinces—Shandong (eastern China), Jiangsu (eastern China), Guizhou (southwestern China), and Hebei (northern China)—were accessible and subsequently selected for analysis. These regions housed subjects with varied lifestyles and economic statuses. A total of 1729 subjects, aged 6–17, with comprehensive records from 47 monitoring sites (5 in major cities, 20 in medium and small cities, 14 in typical rural areas, and 8 in deprived rural areas) were included. The distribution of these sites was proportional to their population contributions. A flowchart detailing the inclusion and exclusion criteria is provided in Figure 1a. Propensity score matching (1:4, caliper = 0.02) results were shown in Figure 1b and Appendix A. The study was conducted in accordance with the Declaration of Helsinki and was approved by the National Institute for Nutrition and Health Ethical Review Board of China CDC (No. 201614). Written informed consents were obtained.

### 2.2. Measurement and Definitions

Information regarding food intake was gathered using a validated food frequency questionnaire, administered by trained investigators and, if necessary, with the assistance of parents. Dietary nutrient intakes, including energy and BCAAs intake, were calculated in accordance with the “China Food Composition Table”. Samples with energy intakes >5000 kcal or <500 kcal [15,16], and with BCAAs beyond the mean ± 3 standard deviations were excluded. Additionally, lifestyle information, which included exposure to secondhand smoke, alcohol consumption, and physical activity, was collected using validated questionnaires. Individuals who engaged in less than 60 min of moderate to vigorous physical activity daily were classified as having insufficient physical activity [17].

Anthropometric parameters were ascertained using standard procedures [18]. Height was gauged utilizing a metal column-type stadiometer (Nantong Yuejian Tice Devices and Materials Co.,Ltd. China)boasting an accuracy of 0.1 cm; weight was determined via an electronic scale (Jinrenchao(Beijing)Technology Co.,Ltd. China) with a 0.05 kg accuracy; waist circumference was measured employing a waist measuring tape (Jiangsu Kongki Commodity Co.,Ltd. China) with 0.1 cm precision; and blood pressure was ascertained through an electronic sphygmomanometer (scale range 0–300 mmHg) (OMRON (Dalian) CO., LTD. China) with a 1 mmHg accuracy. Obesity was defined according to Body Mass Index (BMI) [19], and a standardized reference for children aged 6 and those aged 7–17 was employed to identify individuals with elevated blood pressure [20,21].

Fasting blood samples, collected in quantities of 6 mL from each subject, were subsequently stored at a temperature of −20 °C. The measurement of serum lipids was executed using a direct assay method. Subjects demonstrating serum LDL-C levels equal to or exceeding 3.37 mmol/L were categorized as individuals suffering from hyper-LDL-cholesterolemia [22].

### 2.3. Statistics

The importance of BCAAs in predicting hyper-LDL-cholesterolemia has been revealed through the machine learning LightGBM algorithm. We employed conditional logistic regression analysis to examine the correlation between BCAA intake and the risk of hyper-LDL-cholesterolemia prevalence, as well as to calculate the odds ratio (OR) and its 95% confidence interval associated with an increased risk of hyper-LDL-cholesterolemia prevalence due to elevated BCAA intake. We conducted an interaction analysis to discern whether confounding factors modulate the impact of BCAA intake on the risk of hyper-LDL-cholesterolemia prevalence. To elucidate the dose–response relationship between dietary BCAA intake and the risk of hyper-LDL-cholesterolemia prevalence, we executed both RCS and ROC. In the RCS analysis, we used the median of BCAA intake where OR = 1 as a reference point, testing for the potential of a nonlinear relationship between BCAA intake and hyper-LDL-cholesterolemia prevalence risk. With ROC, we calculated the area under the ROC curve (AUC) for each BCAA and employed the Youden index (specificity + sensitivity − 1) to identify the optimal cut-off value of dietary BCAA intake that most effectively differentiates those with and without hyper-LDL-cholesterolemia. We also computed the population attributable fraction (PAF), representing the proportion of hyper-LDL-cholesterolemia prevalence risk that would decrease if past exposure to BCAA were reduced to a reference nutrient intake [23]. As part of our sensitivity analysis, we re-ran the weighted logistic regression analysis within subgroups categorized by gender, age, BMI, blood pressure, drinking status, and physical activity level. Using mediation Mendelian randomization (MR) analysis to explore the causal effects of exposure factors on outcome factors, mediating variables. The process is executed in two stages: initially, the effect of exposure to BCAA on the mediator LDL-C is calculated, followed by the computation of the impact of the mediator LDL-C on the outcome CVD. We used Stata 18.0 for data analysis and R 4.4.1 for figure analysis, defining a two-sided significance level at *p* < 0.05.

## 3. Results

### 3.1. Characteristics of Participants

In this study, 1729 children and adolescents were screened through the inclusion and exclusion criteria (Figure 1). Table 1 presents the baseline demographic characteristics, health examinations, lifestyle, and nutritional intakes of the children and adolescents who participated in this study. From a total of 1729 subjects, 20.2% (350) exhibited hyper-LDL-cholesterolemia. Among these subjects, 55.6% were aged between 6 and 10 years, 21.1% were obese, and 27.8% had elevated blood pressure. The intake levels of isoleucine, leucine, and valine were recorded at 4.50 g/day, 9.12 g/day, and 6.11 g/day, respectively. These levels were found to be 0.13 SD, 0.14 SD, and 0.15 SD higher in subjects with hyper-LDL-cholesterolemia compared to those in the normal population.

### 3.2. Association Between BCAA Intake and Hyper-LDL-Cholesterolemia Prevalence Risk

The conditional logistic regression analysis revealed a statistically significant positive correlation between dietary BCAA intake and the risk of hyper-LDL-cholesterolemia prevalence among children and adolescents aged 6 to 17 years, as shown in Table 2. For isoleucine, the odds ratio (OR) was 1.08 with a 95% confidence interval (CI) of 1.03 to 1.14 and *p* = 0.003. For leucine, the OR was 1.04 with a 95% CI of 1.01~1.06 and *p* = 0.003. For valine, the OR was 1.06 with a 95% CI of 1.02~1.10 and *p* = 0.002. This correlation persisted even after adjusting for variables such as elevated blood pressure, secondhand smoke exposure, alcohol consumption, lack of physical activity, and energy intake. The adjusted OR for isoleucine was 1.30 with a 95% CI of 1.17~1.45 and *p* < 0.001. For leucine, the adjusted OR was 1.11 with a 95% CI of 1.06~1.17 and *p* < 0.001. For valine, the adjusted OR was 1.16 with a 95% CI of 1.09~1.24 and *p* < 0.001. In essence, the risk of hyper-LDL-cholesterolemia prevalence increased by 30% with each additional 1 g/day intake of isoleucine, by 11% with each additional 1 g/day intake of leucine, and by 16% with each additional 1 g/day intake of valine.

### 3.3. Machine Learning: LightGBM Algorithm

According to the variable importance order and the Shapley value figure, BCAAs, especially isoleucine, play a very important role in the prediction of hyper-LDL-cholesterolemia (Figure 2).

### 3.4. Interaction Analysis

We performed an interaction analysis to determine whether the confounding factors, including high blood pressure, passive smoke exposure, alcohol consumption, and physical inactivity, affected the relationship between BCAA intake and hyper-LDL-cholesterolemia risk (Table 3). We found that none of these factors significantly modified the association between BCAA intake and hyper-LDL-cholesterolemia risk (*p* for interaction > 0.05).

### 3.5. Dose–Response Relationship Analysis

The Restricted Cubic Splines (RCS) were employed to delve deeper into the dose–response relationship between dietary BCAA intake and the risk of hyper-LDL-cholesterolemia prevalence (Figure 3). Median intake values of isoleucine (3.76 g/day), leucine (7.60 g/day), and valine (5.05 g/day) were utilized as reference points. The analysis indicated a linear correlation between BCAA intake and the risk of hyper-LDL-cholesterolemia prevalence, as evidenced by a non-significant *p*-value for nonlinearity (>0.05) and a significant *p*-value for the overall model (<0.05).

Receiver Operator Characteristic (ROC) curves were utilized to ascertain the optimal cut-off values for dietary BCAA intake. As depicted in Figure 4, the areas under the ROC curves (AUC) for isoleucine, leucine, and valine were found to be 0.58. The optimal cut-off value for isoleucine intake, as calculated by Youden’s index, was 5.53 g/day, with a sensitivity of 0.57 and a 1-specificity of 0.45. For leucine intake, the optimal cut-off value was 6.40 g/day, with a sensitivity of 0.58 and a 1-specificity of 0.47. Finally, for valine intake, the optimal cut-off value was 4.18 g/day, with a sensitivity of 0.59 and a 1-specificity of 0.48.

### 3.6. Population Attributable Fraction (PAF)

The recommended daily intake of isoleucine, leucine, and valine was determined by multiplying the estimated average requirement (mg/kg/d) by a coefficient of 1.25 [24]. We assessed the potential reduction in the prevalence risk of hyper-LDL-cholesterolemia if the previous exposure to branched-chain amino acids (BCAA), measured in mg/kg/d, was decreased to the proposed nutrient intake levels for BCAA. This potential reduction, or population attributable fraction (PAF), ranged from 24.00% to 49.22% for isoleucine, 17.78% to 53.14% for leucine, and 18.68% to 53.27% for valine (Figure 5).

### 3.7. Sensitivity Analysis

The results of the subgroup analysis are presented in Table 3. We repeated the conditional logistic regression analysis within subgroups of different genders, ages, BMIs, blood pressures, drinking statuses, and physical activity levels. The correlation of BCAA intake and hyper-LDL-cholesterolemia prevalence risk was best remained in subgroups of those with normal blood pressure (Isoleucine: OR 1.25, 95% CI 1.11~1.42, *p* < 0.001; Leucine: OR 1.09, 95%CI 1.03~1.15, *p* = 0.003; Valine: OR 1.12, 95%CI 1.04~1.21, *p* = 0.002),those with elevated blood pressure (Isoleucine: OR 1.51, 95% CI 1.21~1.89, *p* < 0.001; Leucine: OR 1.22, 95%CI 1.11~1.35, *p* < 0.001; Valine: OR 1.32, 95%CI 1.16~1.50, *p* < 0.001), of those not exposure to secondhand smoke (Isoleucine: OR 1.20, 95% CI 1.04~1.40, *p* = 0.015; Leucine: OR 1.07, 95%CI 1.01~1.15, *p* = 0.033; Valine: OR 1.11, 95%CI 1.02~1.21, *p* = 0.021), of those exposed to secondhand smoke (Isoleucine: OR 1.45, 95% CI 1.24~1.70, *p* < 0.001; Leucine: OR 1.17, 95%CI 1.09~1.25, *p* < 0.001; Valine: OR 1.24, 95%CI 1.13~1.37, *p* < 0.001), of those not drink alcohol (Isoleucine: OR 1.30, 95% CI 1.16~1.46, *p* < 0.001; Leucine: OR 1.11, 95%CI 1.06~1.17, *p* < 0.001; Valine: OR 1.16, 95%CI 1.08~1.24, *p* < 0.001), of those with sufficient physical activity (Isoleucine: OR 1.36, 95% CI 1.11~1.66, *p* = 0.003; Leucine: OR 1.12, 95%CI 1.03~1.21, *p* = 0.007; Valine: OR 1.16, 95%CI 1.04~1.30, *p* = 0.009), and of those lack of physical activity (Isoleucine: OR 1.27, 95%CI 1.12~1.44, *p* < 0.001; Leucine: OR 1.11, 95%CI 1.05~1.18, *p* < 0.001; Valine: OR 1.16, 95%CI 1.08~1.26, *p* < 0.001).

### 3.8. Mendelian Randomization for Mediation Analysis

Most methods supported a significant positive total effect of BCAA on the risk of CVD, of which about 18% were mediated through LDL-C. Additionally, BCAA may directly affect the risk of CVD through other pathways independent of LDL-C. Although there were differences in the proportion mediated by different methods, the results of the inverse variance weighted method and weighted median supported the mediating role of LDL-C (Table 4). Using the inverse variance weighted method to estimate, the total effect of BCAA on CVD was OR = 1.06 (95% CI: 1.02~1.11), *p* = 0.005; the indirect effect mediated by LDL-C was OR = 1.02 (95% CI: 1.00~1.02), *p* = 0.026; the direct effect was OR = 1.05 (95% CI: 1.01~1.09), *p* = 0.017.

## 4. Discussion

The results of this study indicated that dietary BCAAs are linearly positively correlated with hyper-LDL-cholesterolemia in children and adolescents. For every 1 g/d increase in isoleucine, leucine, and valine intake, the risk of hyper-LDL-cholesterolemia increases by 30%, 11%, and 16%, respectively. The optimal BCAA intake cut-off points to distinguish between hyper-LDL-cholesterolemia and non-hyper-LDL-cholesterolemia in children and adolescents were 5.53 g/day for isoleucine, 6.40 g/day for leucine, and 4.18 g/day for valine. BCAAs have a positive effect on CVD through the mediation of LDL-C.

Dietary BCAAs influence lipid metabolism by stimulating mTORC [25,26]. The mTOR pathway can integrate the information of energy and nutrients to guide the growth and metabolism of eukaryotic cells [26,27]. Dietary branched-chain amino acids are mainly obtained from food, especially animal-based foods [5,28]. Common dietary sources of BCAAs include meat, fish, and dairy products. When a high-BCAA diet is consumed, circulating BCAA concentrations also increase [29]. Increased intracellular BCAA levels, especially leucine, promote mTORC1 activation [30]. BCAAs promote the occurrence and development of cardiovascular disease, dependent on TG metabolism via activation of the mTOR [31]. This will further impact the expression of genes involved in lipid synthesis and degradation, leading to elevated LDL-C levels [32]. On the other hand, BCAA may drive adipose tissue inflammation by activating the IFNGR1/JAK1/STAT1 signaling pathway, leading to the polarization of adipose tissue macrophages (ATM) towards a pro-inflammatory (M1) phenotype [33].

A healthy diet helps children and adolescents prevent dyslipidemia and reduce the risk of CVD. Some research indicates that dietary risk factors contribute to 40% of CVD mortality [34]. A study indicated that the regular diet of Poles is high in fats, cholesterol, and proteins, which is more detrimental to health in terms of deficiencies than properly balanced diets that eliminate products of animal origin [35]. A high-protein diet may lead to glomerular hyperfiltration, which in turn affects kidney function [36]. An animal-based saturated fat diet with higher plasma BCAA can increase the risk of cardiometabolic derangements [37]. Laboratory studies have shown that diets lacking BCAAs inhibit lipogenesis and reduce liver triglycerides in murine livers [38]. And short-term dietary BCAA restriction reduces fasting BCAA levels to some extent [39,40], which suggests that reducing the risk of obesity and type 2 diabetes by controlling dietary BCAA intake seems feasible. Our results suggested that a higher risk of hyper-LDL-cholesterolemia was associated with intakes of isoleucine, leucine, and valine above 3.76 g/day, 7.60 g/day, and 5.05 g/day, respectively. Controlling the intake of BCAAs at the recommended levels, the population attributable fractions for isoleucine, leucine, and valine were 24.00%~49.22%, 17.78%~53.14%, and 18.68%~53.27%, respectively. The proportion of hyper-LDL-cholesterolemia prevalence risk that would be reduced through controlling diet.

The strength of our study lies in its focus on the population of children and adolescents, a demographic for which, to the best of our knowledge, no such data analyses have been previously conducted. Furthermore, we determined the optimal cut-off values for BCAA intake based on a dose–response relationship. However, this study also has its limitations. There is a potential for recall bias in the collection of dietary data. Additionally, further evidence is required to ascertain the universality of the correlation between dietary BCAA intake and the risk of hyper-LDL-cholesterolemia across regions with diverse dietary patterns and lifestyles.

## 5. Conclusions

The findings indicate a positive correlation between BCAAs intake and the prevalence risk of hyper-LDL-cholesterolemia in children and adolescents. The results reveal that dietary BCAAs may lead to hyper-LDL-cholesterolemia, which could be beneficial in reducing dyslipidemia in children and adolescents.

## Figures and Tables

**Figure 1 nutrients-17-03280-f001:**
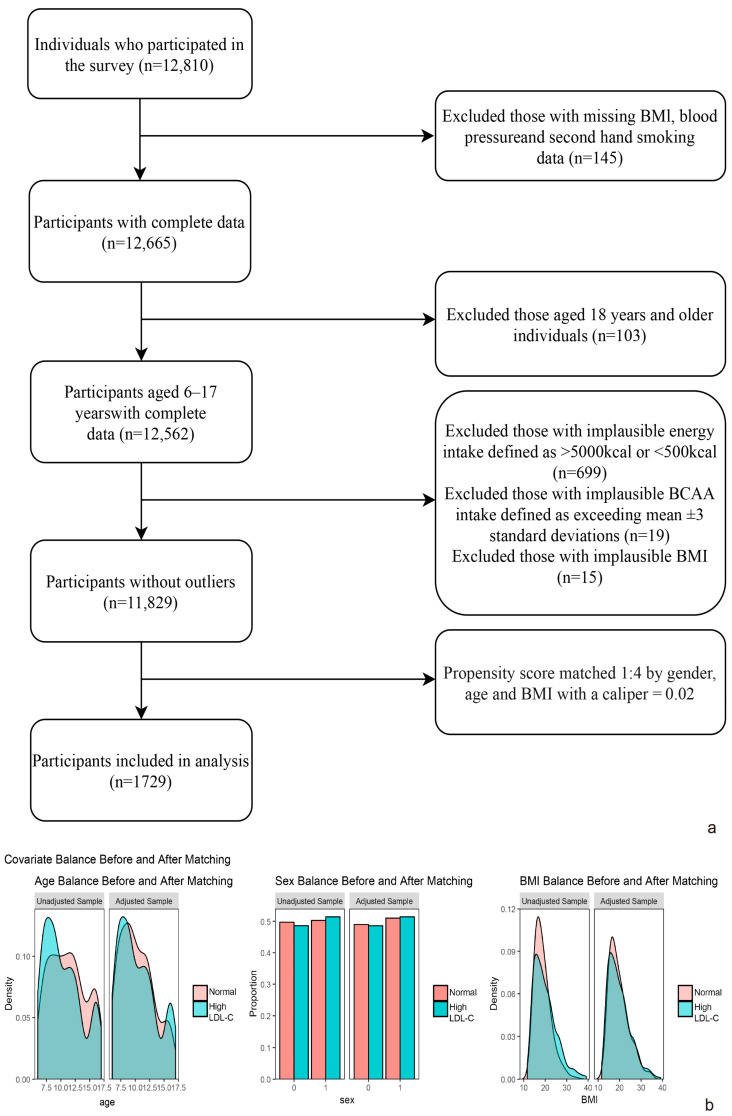
(**a**) Flowchart of inclusion and exclusion; (**b**) propensity score matching covariate balance before and after.

**Figure 2 nutrients-17-03280-f002:**
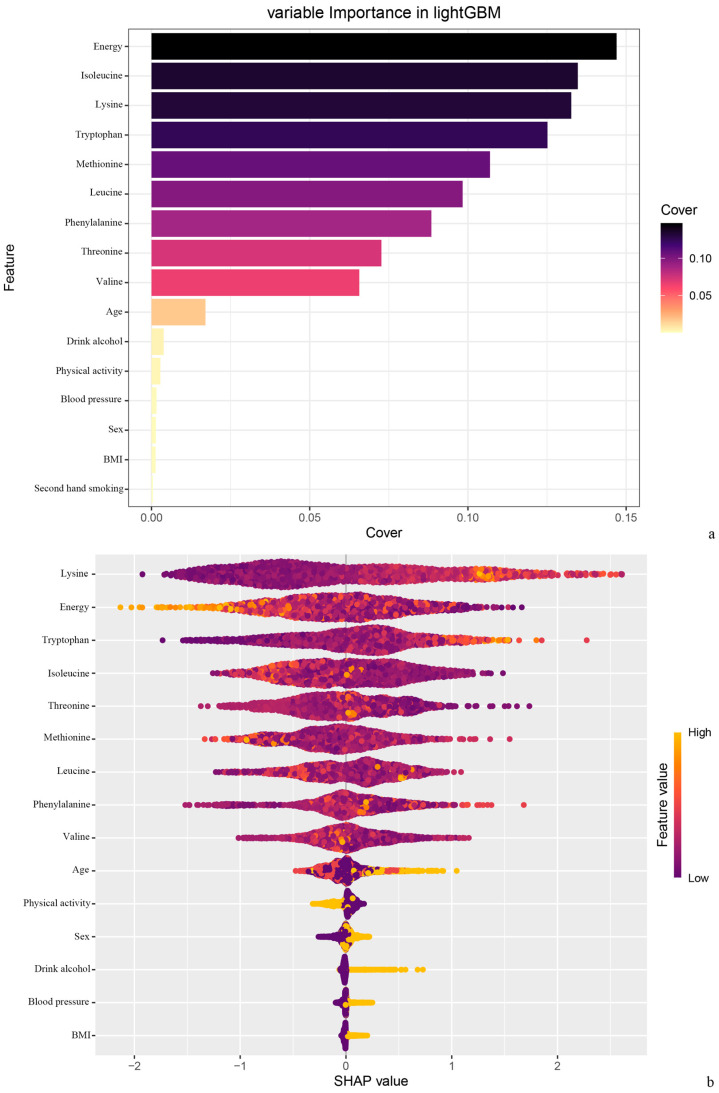
Interpretation and evaluation of machine learning models based on LightGBM. (**a**) Variables importance ranking; (**b**) Influence of variables on predicted results.

**Figure 3 nutrients-17-03280-f003:**
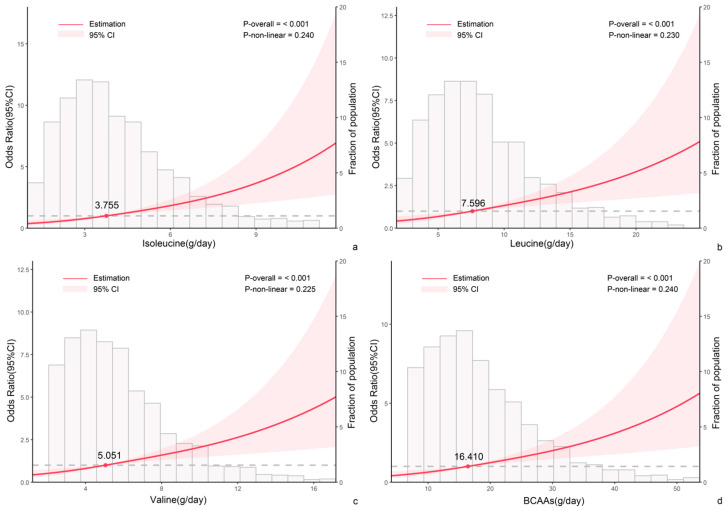
The dose–response relationship between dietary BCAA intake and hyper-LDL-cholesterolemia prevalence risk by Restricted Cubic Splines. (**a**) Isoleucine; (**b**) Leucine; (**c**) Valine; (**d**) BCAAs. OR (hyper-LDL-cholesterolemia prevalence risk) were adjusted for confounding factors, including gender, age, BMI, whether with elevated blood pressure, whether with secondhand smoke, whether with alcohol consumption, whether lack of physical activity, and energy intake (solid red line), and their 95% CI were calculated (the pink area where the solid red line was within). A frequency histogram of BCAA intake was also included (light pink).

**Figure 4 nutrients-17-03280-f004:**
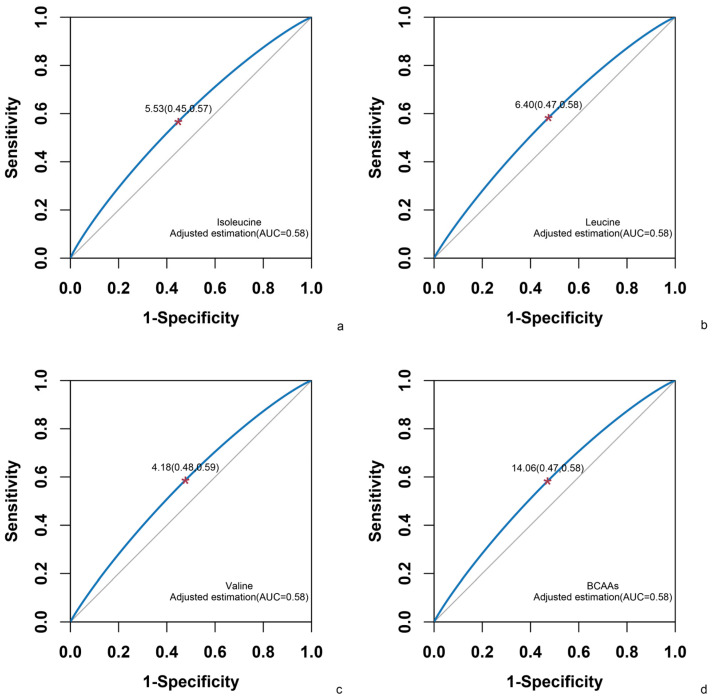
The area under the ROC curve (AUC) for each BCAA and its optimal cut-off values. (**a**) Isoleucine; (**b**) Leucine; (**c**) Valine; (**d**) BCAAs. The model was adjusted for gender, age, BMI, presence of elevated blood pressure, exposure to secondhand smoke, alcohol consumption, lack of physical activity, and energy intake. * denotes the optimal critical points.

**Figure 5 nutrients-17-03280-f005:**
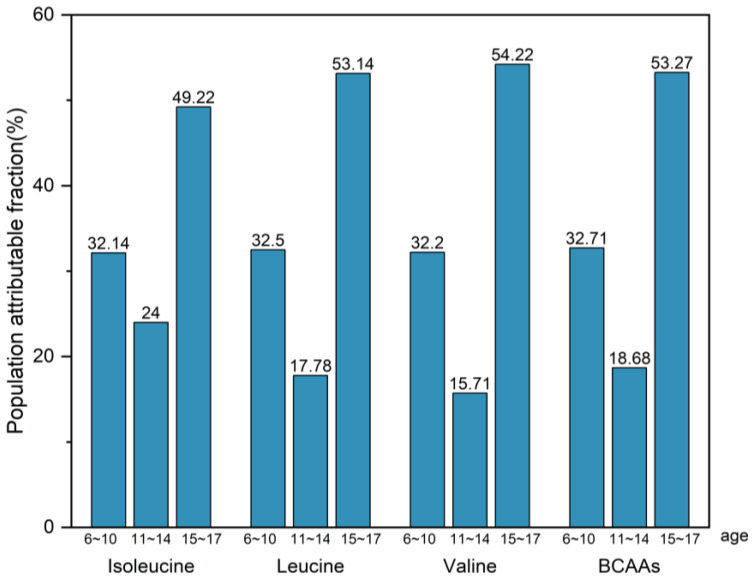
The population attributable fraction (PAF) or the proportion of hyper-LDL-cholesterolemia prevalence risk that would be reduced if the exposure to BCAA in the past were reduced to the reference nutrient intake. The calculation was adjusted for gender, BMI, whether the individual had elevated blood pressure, exposure to secondhand smoke, alcohol consumption, lack of physical activity, and energy intake.

**Table 1 nutrients-17-03280-t001:** Characteristics of children and adolescents in the cross-sectional study.

	Total (n = 1729)		Normal		High Blood LDL-C		χ^2^/t	*p*-Value
	Mean	SD	Mean	SD	Mean	SD		
Gender (n, %)							0.006	0.938
Male	843 (48.8)		673 (38.9)		170 (9.8)			
Female	886 (51.2)		706 (40.8)		180 (10.4)			
Age (n, %)	10.4	2.99	10.39	2.94	10.46	3.20	−0.390	0.697
6~10 years	962 (55.6)		768 (44.4)		194 (11.2)			
11~14 years	545 (31.5)		443 (25.6)		102 (5.9)			
15~17 years	222 (12.8)		168 (9.7)		54 (3.1)			
BMI [kg/m^2^, (n, %)]	19.66	4.61	19.61	4.48	19.85	5.07	−0.826	0.409
Non-obesity	1364 (78.9)		1095 (63.3)		269 (15.6)			
Obesity	365 (21.1)		284 (16.4)		81 (4.7)			
Blood pressure (n, %)							0.047	0.827
Normal	1248 (72.2)		997 (57.7)		251 (14.5)			
Elevated blood pressure	481 (27.8)		382 (22.1)		99 (5.7)			
Secondhand smoking (n, %)							0.004	0.950
No	1035 (59.9)		826 (47.8)		209 (12.1)			
Yes	694 (40.1)		553 (32.0)		141 (8.2)			
Drinker (n, %)							0.645	0.422
No	1589 (91.9)		1271 (73.5)		318 (18.4)			
Yes	140 (8.1)		108 (6.2)		32 (1.9)			
Lack of physical activity (n, %)							0.087	0.768
Yes	1187 (68.7)		949 (54.9)		238 (13.8)			
No	542 (31.3)		430 (24.9)		112 (6.5)			
Energy intake (kcal/day)	1940.39	923.21	1932.48	922.68	1971.56	925.97	−0.707	0.480
Isoleucine intake (g/day)	4.20	2.15	4.12	2.10	4.51	2.32	−2.812	0.005
Leucine intake (g/day)	8.51	4.45	8.35	4.35	9.13	4.77	−2.948	0.003
Valine intake (g/day)	5.68	3.02	5.56	2.95	6.12	3.25	−2.926	0.004
BCAAs intake (g/day)	18.38	9.58	18.03	9.36	19.76	10.31	−3.017	0.003

**Table 2 nutrients-17-03280-t002:** Logistic regression analysis of the association between branched-chain amino acid intake levels and the risk of hyper-LDL-cholesterolemia among children and adolescents aged 6 to 17 years.

	Model 1			Model 2			Model 3		
	OR	95%CI	*p*-Value	OR	95%CI	*p*-Value	OR	95%CI	*p*-Value
Conditional logistic regression model									
Isoleucine	1.08	(1.03, 1.14)	0.003	1.08	(1.03, 1.14)	0.003	1.30	(1.17, 1.45)	<0.001
Leucine	1.04	(1.01, 1.06)	0.003	1.04	(1.01, 1.06)	0.003	1.11	(1.06, 1.17)	<0.001
Valine	1.06	(1.02, 1.10)	0.002	1.06	(1.02, 1.10)	0.002	1.16	(1.09, 1.24)	<0.001
BCAAs	1.02	(1.01, 1.03)	0.003	1.02	(1.01, 1.03)	0.003	1.05	(1.03, 1.08)	<0.001
Logistic regression mixed effects model									
Isoleucine	1.09	(1.02, 1.17)	0.009	1.09	(1.02, 1.16)	0.009	1.34	(1.17, 1.54)	<0.001
Leucine	1.04	(1.01, 1.08)	0.010	1.04	(1.01, 1.08)	0.010	1.13	(1.06, 1.20)	<0.001
Valine	1.07	(1.02, 1.12)	0.007	1.07	(1.02, 1.12)	0.007	1.18	(1.09, 1.29)	<0.001
BCAAs	1.02	(1.01, 1.04)	0.009	1.02	(1.01, 1.04)	0.009	1.06	(1.03, 1.09)	<0.001

Model 1 only included Isoleucine/Leucine/Valine/BCAA total intake as an independent variable. Model 2 adjusted blood pressure in addition to Model 1. Model 3 adjusted confounding factors, including whether with secondhand smoke, whether with alcohol consumption, whether with lack of physical activity, and energy intake, in addition to Model 2.

**Table 3 nutrients-17-03280-t003:** Subgroup analysis of associations between branched-chain amino acid intake levels and the risk of developing hyper-LDL-cholesterolemia.

	Isoleucine		*p* for Interaction	Leucine		*p* for Interaction	Valine		*p* for Interaction	BCAAs		*p* for Interaction
	OR	95%CI		OR	95%CI		OR	95%CI		OR	95%CI	
Blood pressure			0.233			0.408			0.449			0.372
Normal	1.25	(1.11, 1.42)		1.09	(1.03, 1.15)		1.12	(1.04, 1.21)		1.04	(1.02, 1.07)	
Elevated blood pressure	1.51	(1.21, 1.89)		1.22	(1.11, 1.35)		1.32	(1.16, 1.50)		1.10	(1.05, 1.15)	
Secondhand smoking			0.589			0.756			0.763			0.710
No	1.20	(1.04, 1.40)		1.07	(1.01, 1.15)		1.11	(1.02, 1.21)		1.04	(1.01, 1.07)	
Yes	1.45	(1.24, 1.70)		1.17	(1.09, 1.25)		1.24	(1.13, 1.37)		1.08	(1.04, 1.11)	
Drinker			0.884			0.884			0.965			0.903
No	1.30	(1.16, 1.46)		1.11	(1.06, 1.17)		1.16	(1.08, 1.24)		1.05	(1.03, 1.08)	
Yes	1.29	(0.98, 1.69)		1.12	(0.99, 1.26)		1.18	(0.99, 1.41)		1.05	(0.99, 1.12)	
Lack of physical activity			0.847			0.732			0.645			0.722
No	1.36	(1.11, 1.66)		1.12	(1.03, 1.21)		1.16	(1.04, 1.30)		1.06	(1.02, 1.10)	
Yes	1.27	(1.12, 1.44)		1.11	(1.05, 1.18)		1.16	(1.08, 1.26)		1.05	(1.02, 1.08)	

**Table 4 nutrients-17-03280-t004:** Causal association between BCAA and CVD: evidence from Mendelian randomization for mediation analysis.

Method	Total Effect OR (95%CI)	*p*-Value	Indirect Effect OR (95%CI)	*p*-Value	Direct Effect OR (95%CI)	*p*-Value	Mediated Proportion (%)
MR Egger	0.99 (0.92, 1.07)	0.817	0.99 (0.98, 1.01)	0.903	0.99 (0.92, 1.07)	0.829	10.5
Weighted median	1.03 (1.01, 1.05)	0.003	1.01 (1.00, 1.01)	0.003	1.02 (1.00, 1.04)	0.022	24.4
Inverse variance weighted	1.06 (1.02, 1.11)	0.005	1.02 (1.00, 1.02)	0.026	1.05 (1.01, 1.09)	0.017	18.0
Simple mode	1.04 (1.00, 1.09)	0.070	1.02 (1.00, 1.04)	0.042	1.02 (0.98, 1.06)	0.255	48.7
Weighted mode	1.03 (1.01, 1.05)	0.003	1.00 (0.99, 1.01)	0.067	1.03 (1.01, 1.04)	<0.001	13.8

## Data Availability

Restrictions apply to the availability of these data. Data were obtained from our project group and are available upon permission from our project group.

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
