# Peer review of "Dietary Branched-Chain Amino Acids and Hyper-LDL-Cholesterolemia: A Case–Control Study Using Interpretable Machine-Learning Models in Chinese Children and Adolescents"

_nutrients, 2025, doi:10.3390/nu17203280_

Round 1
Reviewer 1 Report
Comments and Suggestions for Authors
The problem of hyper-LDL cholesterolemia is mainly investigated in the context of adults. Children and adolescents have been less investigated. From the medical point of view, it is a forgotten mistake. The health problem in the early years of life will have a negative effect in older age. Therefore, the article entitled Dietary Branched-Chain Amino Acids and Hyper-LDL-2 Cholesterolemia: A Case-Control Study Using Interpretable 3 Machine-Learning Models in Chinese Children and 4 Adolescents is at the central point of the medical health problem. The authors in their studies have brought into consideration the consumption and level of branched-chain amino acids, which level is well correlated with the fraction of LDL cholesterol. Moreover, the discussed amino acids are present in the meat-based nutrition. The above is typical of fast and highly processed food, which is certainly prevalent in the environment of young people.
The methodology, bioinformatics, and machine learning, in fact, are correctly selected and described. Moreover, the investigated population is sufficient to give statistically significant results. I agree with the author that the food consumption described by probants poses some uncertainty (is not objective).
The article is well written and readable with correctly selected literaturÄ™. From the graphical point of view, I recommend moving the figures to supplementary materials and increasing their resolution.
Author Response
Reviewer 1:
The problem of hyper-LDL cholesterolemia is mainly investigated in the context of adults. Children and adolescents have been less investigated. From the medical point of view, it is a forgotten mistake. The health problem in the early years of life will have a negative effect in older age. Therefore, the article entitled Dietary Branched-Chain Amino Acids and Hyper-LDL-2 Cholesterolemia: A Case-Control Study Using Interpretable 3 Machine-Learning Models in Chinese Children and 4 Adolescents is at the central point of the medical health problem. The authors in their studies have brought into consideration the consumption and level of branched-chain amino acids, which level is well correlated with the fraction of LDL cholesterol. Moreover, the discussed amino acids are present in the meat-based nutrition. The above is typical of fast and highly processed food, which is certainly prevalent in the environment of young people.
The methodology, bioinformatics, and machine learning, in fact, are correctly selected and described. Moreover, the investigated population is sufficient to give statistically significant results. I agree with the author that the food consumption described by probants poses some uncertainty (is not objective).
The article is well written and readable with correctly selected literaturÄ™. From the graphical point of view, I recommend moving the figures to supplementary materials and increasing their resolution.
Reply to the reviewer 1:
Thank you very much for your comments and suggestion. We consider the figures should be kept in the main text, as it is beneficial to presenting the results and reading. We have enhanced the resolution of the figures, and the pixels meet the requirements. Thanks again.
Reviewer 2 Report
Comments and Suggestions for Authors
Points for consideration:
1). Line 56: should be changed to 'research has demonstrated ...'.
2). Line 57: discuss the 'negative' connotations of BCAA in a more balanced light, why is it bad when it is essential to our bodies, and we cannot make it? Is the correlation here simply due to the fact that with more BCAAs through the diet, naturally more red meat is consumed, which also means more lipids and ROS? Simply stating that BCAAs are bad is simplistic and even dangerous. Provide a more rationale even-keeled dialogue, not one steeped with nuances of varying bias.
3). Line 75: define what NINH stands for, the long name.
4). Line 82: it would have been interesting to seek out the effects of dietary total protein intake on measurements outcomes too, not just the BCAA ones.
5). Table 1: hard to follow, as not easy to discern which LDL-C belongs to which BCAA reading.
6). Line 271: not a full sentence.
Author Response
Reply to the reviewer 2:
Reviewer 2:
Points for consideration:
1). Line 56: should be changed to 'research has demonstrated ...'.
Answer to the reviewer: Thanks for your advice. We have changed the original sentence to "Current research has demonstrated that serum BCAA contribute to the development of CVD". (Line 55)
2). Line 57: discuss the 'negative' connotations of BCAA in a more balanced light, why is it bad when it is essential to our bodies, and we cannot make it? Is the correlation here simply due to the fact that with more BCAAs through the diet, naturally more red meat is consumed, which also means more lipids and ROS? Simply stating that BCAAs are bad is simplistic and even dangerous. Provide a more rationale even-keeled dialogue, not one steeped with nuances of varying bias.
Answer to the reviewer: Thank you for pointing this out. We agree with this comment. The increase in BCAAs intake may lead to an increase in the consumption of foods such as red meat, resulting in higher lipid intake. This is a concern worth noting, but what we should pay more attention to is the increase in BCAAs intake due to diet. BCAA has both benefits and harms to the human body, and we agree to elaborate from a more balanced perspective. The relevant elaboration has been modified in the second paragraph of the introduction section (Line 56).
3). Line 75: define what NINH stands for, the long name.
Answer to the reviewer: Thank you for pointing this out. We have defined in the original (National Institute for Nutrition and Health). (Line 79)
4). Line 82: it would have been interesting to seek out the effects of dietary total protein intake on measurements outcomes too, not just the BCAA ones.
Answer to the reviewer: Thanks for your valuable question. The protein intake in fact is in the same direction as BCAA. However, due to the potential collinearity between protein intake and BCAA, it is not included in the analysis.
5). Table 1: hard to follow, as not easy to discern which LDL-C belongs to which BCAA reading.
Answer to the reviewer: Thanks for your suggestion. We adjusted the column width of Table 1 for easier reading.
6). Line 271: not a full sentence.
Answer to the reviewer: We sincerely thank you for the careful reading. We have modified it at line 284. Thanks again.
Reviewer 3 Report
Comments and Suggestions for Authors
The authors analyze the association between branched-chain amino acid (BCAA) intake and the risk of LDL hypercholesterolemia, using both classical epidemiological analysis methods (conditional regression, restricted cubic splines) and modern machine learning tools (LightGBM), as well as mediation analysis based on Mendelian randomization. The results indicate a positive, linear association between higher BCAA intake and the risk of elevated LDL-C, as well as an indirect effect of this factor on the risk of cardiovascular disease.
1. The introduction is very short and does not adequately inform the reader about the current state of knowledge.
2. There is also no description of the type of diet as a potential source of BCAAs (NUTRIENTS journal).
3. An explanation of abbreviations is required upon first use in both the abstract and the manuscript.
4. There is no explanation of mTOR and the role of this pathway.
Establishing thresholds for BCAA intake may have potential applications in the prevention of dyslipidemia in children and adolescents. 4. The discussion requires the introduction of products rich in isoleucine, leucine, and valine, and the specific amounts of these products that can disrupt this balance.
5. The mTOR pathway and its activation are not fully understood.
6. The discussion should consider which other dietary factors activate mTOR and which of them are most important.
7. Can you identify the factors that play the most important role in mTOR pathway activation?
8. What other pathways are activated?
doi: 10.1186/s10020-024-00894-9. 9. The sentence "...And short-term dietary BCAA restriction reduces fasting BCAA levels to some extent" [33,34] seems unclear and unfinished.
10. Please specify what specifically controls the intake of isoleucine, leucine, and valine.
DOI: 10.3390/nu12102986
doi: 10.1016/j.cmet.2021.03.025
11. It is suggested to discuss the following issues:
- A high protein intake poses a risk of excessive strain on the kidneys and their dysfunction.
- Too high a protein intake in the diet without increasing physical activity necessary for muscle growth may reduce bone density by lowering pH.
- A comparison of the risks of various diets from different regions.
DOI: 10.3390/nu12102986
12. The text is written correctly, but there are minor language corrections in several places. (Shortening overly long sentences, especially in the materials and methods sections, and avoiding repetitions) could improve the flow and comprehension of the article.
At this stage, the article is not suitable for publication.
Author Response
Reply to the reviewer 3:
Reviewer 3:
The authors analyze the association between branched-chain amino acid (BCAA) intake and the risk of LDL hypercholesterolemia, using both classical epidemiological analysis methods (conditional regression, restricted cubic splines) and modern machine learning tools (LightGBM), as well as mediation analysis based on Mendelian randomization. The results indicate a positive, linear association between higher BCAA intake and the risk of elevated LDL-C, as well as an indirect effect of this factor on the risk of cardiovascular disease.
- The introduction is very short and does not adequately inform the reader about the current state of knowledge.
Answer to the reviewer: Thanks for your suggestion. We added relevant expressions of BCAAs in the introduction section.
- There is also no description of the type of diet as a potential source of BCAAs (NUTRIENTS journal).
Answer to the reviewer: Thank you for pointing this out. We indicated in the Introduction (line 50) that the main food source of dietary BCAA is animal-based proteins such as red meat. And we added meat, fish, and dairy products as common dietary sources of BCAAs (line 268).
- An explanation of abbreviations is required upon first use in both the abstract and the manuscript.
Answer to the reviewer: Thanks for your careful reading. We explained the abbreviations when they first use in the manuscript.
- There is no explanation of mTOR and the role of this pathway.
Establishing thresholds for BCAA intake may have potential applications in the prevention of dyslipidemia in children and adolescents. 4. The discussion requires the introduction of products rich in isoleucine, leucine, and valine, and the specific amounts of these products that can disrupt this balance.
Answer to the reviewer: Thanks for your comments. We added the explanation about mTOR pathway in the Discussion section (line 266). We added meat, fish, and dairy products as common dietary sources of BCAAs (line 268). There is no exact standard for the specific amounts of these products. And our results suggested that a higher risk of hyper-LDL-cholesterolemia was associated with isoleucine, leucine and valine intakes above 3.76 g/day, 7.60 g/day and 5.05 g/day per day, respectively (line 289).
- The mTOR pathway and its activation are not fully understood.
Answer to the reviewer: Thanks for your commnts. mTOR is a catalytic subunit of two structurally and functionally distinct complexes, mTORC1 and mTORC2. Increased intracellular levels of amino acids and especially leucine, promote the activation of mTORC1 through the active complex of Rag guanosine triphosphatases (GTPases) consisting of GTP-bound RagA/B and GDP-bound RagC/D.
- The discussion should consider which other dietary factors activate mTOR and which of them are most important.
Answer to the reviewer: Thanks for your suggestion. In terms of amino acids, leucine is considered the most important activator of mTORC1. And other dietary factors such as glucose and fatty acids can also activate mTOR. We mentioned in line 271 that leucine is the most important activation of mTORC1.
- Can you identify the factors that play the most important role in mTOR pathway activation?
Answer to the reviewer: Thank you for pointing this question. Growth factors, nutrients, energy and oxygen availability, and other cellular stresses are all important factors in regulating mTOR, with amino acids playing a dominant role as far as mTORC1 is concerned. DOI: 10.1038/s41580-020-0219-y
- What other pathways are activated?
doi: 10.1186/s10020-024-00894-9.
Answer to the reviewer: Thanks for your valuable suggestion. As you mentioned the ref, BCAA may drive adipose tissue inflammation by activating IFNGR1/JAK1/STAT1 signaling to polarize ATMs toward a proinflammatory (M1) phenotype. This is distinct from the mTOR pathway, which we added at line 274.
- The sentence "...And short-term dietary BCAA restriction reduces fasting BCAA levels to some extent" [33,34] seems unclear and unfinished.
Answer to the reviewer: Thanks for your careful reading. We have revised the relevant description (line 286).
- Please specify what specifically controls the intake of isoleucine, leucine, and valine.
DOI: 10.3390/nu12102986doi: 10.1016/j.cmet.2021.03.025
Answer to the reviewer: Thanks for your suggestion. Reducing meat, fish and dairy products intake would lead to a significant reduction in BCAAs intake. This is consistent with what we have mentioned above that the main food sources of BCAAs are meat, fish and dairy products.
- It is suggested to discuss the following issues:
- A high protein intake poses a risk of excessive strain on the kidneys and their dysfunction.
- Too high a protein intake in the diet without increasing physical activity necessary for muscle growth may reduce bone density by lowering pH.
- A comparison of the risks of various diets from different regions.
DOI: 10.3390/nu12102986IF: 5.0 Q1
Answer to the reviewer: Thanks for your valuable advice. We added the related content that high-protein diet may have damage to the kidney function and the risks of various diets from different regions in the Discussion (line 280).
- The text is written correctly, but there are minor language corrections in several places. (Shortening overly long sentences, especially in the materials and methods sections, and avoiding repetitions) could improve the flow and comprehension of the article.
At this stage, the article is not suitable for publication.
Answer to the reviewer: Thanks for your careful check. We made appropriate modifications to the Materials and Methods section. Thank you again.
Round 2
Reviewer 3 Report
Comments and Suggestions for Authors
The authors responded very promptly to all suggestions. I have no further questions, and congratulations.
best regards